# OpenReview forum: "Generative Adaptation of Dynamics to Environmental Shifts via Weight-space Diffusion"
_ICML.cc/2026/Conference — ICML 2026 regular_

### Official Review · Reviewer_sf5a · 2026-03-07

**Soundness:** 3
**Presentation:** 3
**Significance:** 3
**Originality:** 3
**Overall Recommendation:** 4
**Confidence:** 2

**Summary:**

The paper proposes DynaDiff, a novel generative meta-learning framework designed to adapt dynamical systems prediction models to varying environmental conditions. Instead of traditional gradient-based fine-tuning, DynaDiff directly generates the complete weights of a target model by modeling the conditional distribution of weights given environmental observations. The framework represents model weights as "weight graphs" and employs a diffusion model conditioned by a "dynamics-informed prompter" that extracts physical and spectral features from short observation sequences. By pre-constructing a model zoo of expert weights, the approach amortizes the adaptation cost into a one-time offline process, enabling efficient deployment in data-scarce or hardware-constrained scenarios.

**Compliance With Llm Reviewing Policy:**

Affirmed.

**Final Justification:**

Concerns are solved

**Key Questions For Authors:**

see weakness

**Limitations:**

see weakness

**Strengths And Weaknesses:**

Strength:
+ By abstracting weights as weight graphs using multi-head attention, the model captures the topological coupling within neural architectures rather than treating weights as flat vectors.
+ The method achieves a significant 10.78% improvement in prediction accuracy over competitive baselines.


Weakness:
Can other weight space learning approches be used to talckle the tasks in this paper? It is highly recommended to conduct some experiments on this.

---

> ### Author Rebuttal · Authors · 2026-03-31
>
> We thank you for the constructive suggestions. We appreciate your point that verifying the performance of other weight-space learning methods in dynamics adaptation tasks is crucial for establishing the standing of this study.
>
> While most existing weight generation methods are not designed for generalization problems in the Scientific Machine Learning (SciML) field, their core architectures can serve as strong baselines for comparison. To this end, we have added the following two representative methods:
>
> - **CVAE-based Approach [1]**: In response to other reviewers' suggestions regarding the "necessity of non-diffusion methods," we implemented the Conditional Variational Autoencoder from [2]. To ensure absolute fairness in comparison, the network architecture of the CVAE is kept consistent with DynaDiff's Graph VAE. We increased the latent dimension to 1440 to ensure the parameter count is close to DynaDiff.
> - **D2NWG (ICLR 2025) [2]**: Chosen as the state-of-the-art diffusion-based weight generation baseline due to its task setting being closest to DynaDiff (both use Latent Diffusion as the core model).
>
> All models use $L=10$ observation frames from the new environment as the prompt. Experimental results are shown in the table below:
>
> | RMSE     | Params | CF              |                 | LO              |                 | KF              |                 | NS              |                 |
> | -------- | ------ | --------------- | --------------- | --------------- | --------------- | --------------- | --------------- | --------------- | --------------- |
> |          |        | in              | out             | in              | out             | in              | out             | in              | out             |
> | CVAE     | ~390M  | 0.158±0.009     | 0.157±0.009     | 0.486±0.019     | 0.487±0.019     | 4.454±0.128     | 29.089±0.561    | 0.204±0.008     | 0.233±0.014     |
> | D2NWG    | ~400M  | 0.082±0.019     | 0.086±0.017     | 0.102±0.020     | 0.105±0.021     | 0.086±0.010     | 0.090±0.013     | 0.088±0.016     | 0.089±0.015     |
> | DynaDiff | ~380M  | **0.063±0.023** | **0.065±0.021** | **0.088±0.013** | **0.091±0.015** | **0.077±0.008** | **0.079±0.011** | **0.060±0.013** | **0.064±0.012** |
>
> **Result Analysis:**
>
> - Experimental results demonstrate that DynaDiff significantly outperforms existing weight generation methods across all systems.
> - Diffusion-based methods (D2NWG and DynaDiff) generally perform more robustly than the CVAE architecture, especially when dealing with high-dimensional parameter spaces characterized by multimodal distributions.
> - We visualized the joint distribution of weights and environments generated by the three methods and introduced Jensen-Shannon Divergence (JSD) for quantitative analysis ([figure](https://anonymous.4open.science/r/icml-rebuttal-86BD/vis_weight.png)). The results confirm that DynaDiff best fits the true expert weight manifold, with its JSD significantly lower than other baselines.
>
> This further proves the critical roles of our originally proposed weight graph , dynamics-informed prompter, and functional loss  in capturing the essential characteristics of physical dynamics.
>
> ---
>
> We hope these additional experiments and analyses address your concerns. We are happy to discuss further if you have any additional suggestions.
>
>
>
> [1] In-context meta LoRA generation
>
> [2] Diffusion-Based Neural Network Weights Generation

---

> > ### Author Rebuttal · Reviewer_sf5a · 2026-04-02
> >
> > Thanks.

---

> > > ### Author Response · Authors · 2026-04-04
> > >
> > > Thank you for confirming that our responses and the newly added experimental comparisons with weight-space learning baselines addressed your concerns. We appreciate your valuable suggestions that helped strengthen the evaluation of our work.

---

### Official Review · Reviewer_G1JH · 2026-03-12

**Soundness:** 3
**Presentation:** 4
**Significance:** 3
**Originality:** 2
**Overall Recommendation:** 3
**Confidence:** 3

**Summary:**

This paper proposes DynaDiff, a generative adaptation framework for spatiotemporal dynamics forecasting under distribution shifts. Instead of gradient-based fine-tuning at test time, the method learns a conditional distribution over entire model weights given a short observation sequence, and generates an adapted expert model by sampling in weight space. Concretely, the paper (i) introduces a weight-graph representation where nodes correspond to output neurons/channels with node features aggregating incoming weights and biases, (ii) trains a weight VAE (with an additional functional loss encouraging output-level equivalence), and (iii) performs conditional latent diffusion guided by a dynamics-informed prompter extracted from short sequences. Experiments across simulated and real-world settings claim improved generalization to unseen environments/regions and faster adaptation compared to gradient-based baselines.

**Compliance With Llm Reviewing Policy:**

Affirmed.

**Final Justification:**

I think this paper can be accepted though I do think this is really a borderline paper. It should be ranked among the weaker papers accepted by icml

**Key Questions For Authors:**

1. Closest prior work: What is the most similar existing weight-space generative approach to DynaDiff, and what is the clearest technical difference beyond the application? If the difference is substantial, I would raise my originality assessment; if not, I would view the contribution as mainly incremental.

2. Weight graph necessity: Under matched compute/parameters, how does your weight graph compare to simpler alternatives like per-layer structured flattening or a standard hypernetwork generator? If weight graph clearly wins, it strengthens the architectural justification; if not, it weakens it.

3. Symmetry/initialization robustness: How sensitive are results to neuron/channel permutations or different expert-zoo initializations for the same environment? Robustness would increase my confidence in soundness; strong sensitivity would be a serious concern.

4. Practical cost trade-off: Please report end-to-end costs (zoo training, VAE+diffusion training, test-time sampling steps/latency) and compare to a fixed small-step fine-tuning baseline. If adaptation is clearly cheaper/faster at similar accuracy, significance increases; otherwise it decreases.

**Limitations:**

yes

**Strengths And Weaknesses:**

Strengths
Soundness: The technical pipeline is coherent: learn a compact latent space for weights with a weight VAE, then train a conditional latent diffusion model to generate adapted weights from a short observation-derived prompt, avoiding gradient-based test-time fine-tuning. Adding a functional loss to align model behavior (not just parameter values) is an appropriate choice for weight-space learning and helps mitigate non-identifiability issues.

Presentation: The overall narrative is easy to follow and the key constructs are concretely specified, including how the weight graph is built (nodes as output neurons/channels, node features aggregating incoming weights and bias, and how skip parameters are incorporated) and how the prompt is formed and used as conditioning.

Significance: Fast adaptation under distribution shift for spatiotemporal dynamics forecasting is an important problem with real deployment constraints. Framing adaptation as conditional weight generation from short sequences is potentially useful beyond the specific benchmarks, and could inspire similar approaches in other settings where environment-to-weights mappings are natural.

Originality: The work’s originality is mainly in a domain-tailored combination: a structured weight tokenization (weight graph) instead of naive flattening, behavior-level (functional) constraints during weight autoencoding, and conditioning via a dynamics-informed prompt from short sequences when explicit environment identifiers may be unavailable.

Weaknesses
Soundness: The weight graph construction does not clearly guarantee invariance/equivariance to parameter symmetries (e.g., neuron permutations), so some gains may depend on consistent parameterization rather than function-level structure. The practical compute tradeoff is also unclear: diffusion sampling at inference has nontrivial cost, and the paper would be stronger with explicit latency and compute comparisons against modest fine-tuning budgets. It is also not always fully clear which results rely on optional auxiliary supervision for the prompter versus fully label-free conditioning.

Presentation: Related work positioning against the closest weight-space generative modeling and hypernetwork/meta-learning literature could be sharper, since readers may otherwise view the contribution as incremental. Reproducibility would improve with a more centralized description of prompt features per dataset, inference-time diffusion steps, and the exact procedure for training/selecting the expert model zoo.

Significance: The impact depends on whether the approach consistently beats strong lightweight adaptation alternatives under matched compute and data budgets. The method also assumes a feasible expert-zoo training stage and that short sequences contain enough information to identify the regime; if these assumptions fail, applicability may narrow.

Originality: Most components (autoencoding weights, diffusion denoising in latent space, conditional generation, functional matching) are established ideas, so novelty is best viewed as integration and engineering for this domain. If simpler representations or conditioning schemes achieve similar performance, the originality would feel limited, so stronger ablations are needed to justify the specific design choices.

---

> ### Author Rebuttal · Authors · 2026-03-31
>
> We appreciate the reviewer’s professional assessment and constructive suggestions. We address each points below.
>
> **W1 (Soundness):**
>
> - **Function-level Structure**: Weight graph ensures that DynaDiff can capture the distribution of functionalities across the layers of a neural network ([CF](https://anonymous.4open.science/r/icml-rebuttal-86BD/cy_attn_layers.png)、[LO](https://anonymous.4open.science/r/icml-rebuttal-86BD/lo_attn_layers.png) systems); see the response to **Reviewer RMvc W2**. The ablation study in Appendix Table 9 shows that, under the same Model Zoo, the VAE based on the weight graph significantly outperforms the VAE that directly flattens the structure. This proves that the performance gain originates from capturing function-level structure rather than simple parameterization consistency.
> - **Computational Trade-off**: We compared the fine-tuning costs required for meta-learning baselines (e.g., GEPS, CAMEL) to reach the same accuracy (see **Table 2 in the response to pAWm W3**). Results show that at the similar predictive performance, DynaDiff’s test-time speed is approximately 200% faster than the baselines, greatly alleviating inference pressure in hardware-constrained scenarios.
> - **Auxiliary Supervision**: Only Appendix Section I.6 utilized the auxiliary supervision to demonstrate the interpretability improvement.
>
> **W2 & Q1 & W4 (Presentation):**
>
> - **Related Work**: Regarding your inquiry on technical differences, the following table compares DynaDiff with 7 recent closely related weight-space generation works. The primary differences of DynaDiff include: the **weight graph**, the **dynamics-informed prompter**, and the **function loss**. To highlight our contribution, we added two of the most similar weight generation methods as baselines (see **Reviewer sf5a W1**). The figure ([weight_dist](https://anonymous.4open.science/r/icml-rebuttal-86BD/vis_weight.png)) also visually demonstrates the contributions of our three innovations to fitting the weight distribution compared to methods that directly process flattened vectors (e.g., D2NWG [6]).
>
>   See the [table (click)](https://anonymous.4open.science/r/icml-rebuttal-86BD/related_work.png) for a detailed comparison.
>
> - **Detailed Settings**: During testing, the first $L=10$ frames of each new environment trajectory are selected as prompt features. 1000 diffusion steps are used during inference. Model zoo training details are provided in Appendix Table 3, following the principle of ensuring expert model convergence.
>
> **W3 & Q4 (Significance):**
>
> - **Cost Advantage**: We reported the end-to-end cost and comparison with fine-tuning baselines (see analysis for **Reviewer pAWm W3**); DynaDiff’s adaptation efficiency is much higher than existing schemes, making it highly suitable for low-resource inference.
>
> - **Short Sequence Assumption**: Appendix Table 5 proves that a short sequence with $L=10$ already contains sufficient information to identify the dynamical state. This is a common assumption in dynamical system prediction, i.e., extracting features from historical trajectories.
>
> **Q2**: This is an excellent suggestion. We compared the layer-wise flattening approach in Appendix Table 9 and newly added the CVAE generator baseline (see **Reviewer sf5a W1**). These results prove the necessity of the weight graph.
>
> **Q3**: Regarding your concern about permutation sensitivity, we conducted a permutation experiment. Taking the CF and LO systems as examples, we randomly select 10% of the neurons in random layers from the existing Model Zoo and swap their order and the weight matrices with the preceding and following layers during training.
>
> | Out-domain RMSE        | CY          | LO          |
> | ---------------------- | ----------- | ----------- |
> | DynaDiff + Permutation | 0.068±0.027 | 0.090±0.013 |
> | DynaDiff               | 0.065±0.021 | 0.091±0.015 |
>
> Results show that perturbed weights did not significantly degrade performance. The visualization ([figure](https://anonymous.4open.science/r/icml-rebuttal-86BD/cy_attn_permute_5.png)) indicates the global attention mechanism recognizes a consistent hierarchical scale structure, while intra-layer node scores automatically adapt to neuron permutations. This proves DynaDiff is highly robust to equivalent weight permutations.
>
> ---
>
> [1] Drag-and-Drop LLMs: Zero-Shot Prompt-to-Weights
>
> [2] LoRA-Gen: Specializing Large Language Model via Online LoRA Generation
>
> [3] In-context meta LoRA generation
>
> [4] Continual Adaptation: Environment-Conditional Parameter Generation for Object Detection in Dynamic Scenarios
>
> [5] Recurrent Diffusion for Large-Scale Parameter Generation
>
> [6] Diffusion-Based Neural Network Weights Generation
>
> [7] Text2Weight: Bridging Natural Language and Neural Network Weight Spaces

---

> > ### Author Rebuttal · Reviewer_G1JH · 2026-04-03
> >
> > Thanks authors for submitting rebuttal. I am raising my score.

---

> > > ### Author Response · Authors · 2026-04-04
> > >
> > > Thank you for your constructive feedback and for raising the score. We are pleased that our additional experiments resolved your concerns. We appreciate the time you invested in improving our paper.
> > >
> > > ---update---
> > >
> > > Thank you again for your valuable support and for deciding to raise our score. As the discussion period closes today, we just wanted to send a gentle reminder in case the system hasn't properly registered the updated score yet. We appreciate your time and support.

---

### Official Review · Reviewer_pAWm · 2026-03-12

**Soundness:** 3
**Presentation:** 3
**Significance:** 3
**Originality:** 3
**Overall Recommendation:** 4
**Confidence:** 3

**Summary:**

DynaDiff addresses physics dynamics forecasting under environmental shifts by generating environment-specific model weights from a short test-time observation window, rather than relying on gradient-based finetuning or activation conditioning. Given a brief sequence X_L from a new environment, it learns P(θ|X_L) and samples adapted parameters θ_new for deployment.

To enable weight generation, DynaDiff converts network parameters into a weight graph and compresses it with an attention-based VAE, trained with a functional reconstruction loss that matches model outputs instead of only minimizing parameter-space error. A conditional diffusion model is then trained in the latent space, guided by a dynamics prompter that extracts both physical statistics and spectral-temporal features from X_L.

The approach is trained using a model zoo of environment-specific experts built by pretraining a base model and finetuning per environment. Experiments on multiple PDE benchmarks and ERA5 data show improved in-domain and out-of-domain performance, demonstrating that a large offline generator can produce lightweight adapted predictors for unseen environments.

**Compliance With Llm Reviewing Policy:**

Affirmed.

**Final Justification:**

Taking into account the manuscript and the authors’ rebuttal, I lean toward a Weak Accept.

**Key Questions For Authors:**

1) What is the end-to-end compute cost to build the expert zoo (per environment and total), and how does it compare to training/finetuning baselines under the same budget? A clear breakdown could affect my view of practical significance.

2) Diffusion sampling can produce variability—how stable are generated predictors across different random seeds, and do you observe any unstable/degenerate samples? Reporting variance or a selection strategy would strengthen soundness.

**Limitations:**

yes

**Strengths And Weaknesses:**

## Strengths

• **Soundness**: The problem is clearly defined: adapt to unseen environments from a short context window by generating a new predictor, without test-time finetuning. The method pipeline is coherent (weight representation → latent model → conditional generation), and the ablations generally support that these choices matter. The paper also discusses the offline training vs. online adaptation cost and provides basic efficiency comparisons.

• **Presentation**: The paper is well structured and readable, with a clear end-to-end story from training experts to test-time adaptation. Figures and ablations make it easy to see what components contribute to performance. Overall, the writing is clear enough for an ML audience.

• **Significance**: Environmental shift is an important setting in scientific forecasting, and avoiding per-environment finetuning could be practically useful. Results across multiple PDE datasets and ERA5 suggest the approach is not limited to a single benchmark. The reported gains look meaningful in several out-of-domain settings.

• **Originality**: The main novelty is combining short-trajectory conditioning with weight generation for cross-environment adaptation, using a structured weight representation. While related ideas exist (meta-learning, weight-generation, diffusion), the paper's packaging is tailored to dynamics and appears reasonably distinct. The contribution is more in the system design than in a new standalone component.

## Weaknesses

• **Soundness**: The approach depends heavily on an offline "expert zoo," which may be costly or hard to scale when environments are continuous or numerous. It is unclear how robust the method is when the test environment is far outside what the zoo covers. Evaluation focuses on point accuracy; sampling stability and failure cases are not well analyzed, and comparisons to closely related weight-generation baselines could be broader.

• **Presentation**: Some details that likely affect results are still underspecified (exact weight-graph construction, expert training protocol, and compute/hyperparameters).

• **Significance**: The paper argues for faster adaptation, but an end-to-end cost breakdown (training zoo + generation time) versus simpler alternatives is limited. The demonstrated benefits may be strongest for PDE-like benchmarks with structured shifts. Generalization to noisier or partially observed settings is not established.

• **Originality**: The core idea is close to existing meta-learning/hypernetwork-style conditioning, with diffusion as the generator. The paper could better justify why diffusion is needed over simpler conditional generators and what unique benefits it delivers. Novelty should be framed as a strong combination of known tools rather than a fundamentally new paradigm.

---

> ### Author Rebuttal · Authors · 2026-03-31
>
> We thank you for the constructive feedback and insightful comments regarding our submission.
>
> **W1 (Soundness):**
>
> - **Scalability**:  Environments in our experiments are indeed continuous and numerous. We added robustness experiments regarding the Zoo Size in Appendix Figure 8a/b. Results show that DynaDiff maintains stable performance (fluctuations of less than 10%) even when the size is reduced to $25$.
> - **OOD Robustness**: Experiments in Appendix Section I.2 demonstrate that DynaDiff's performance in extreme OOD scenarios is more robust than existing methods.
> - **Sampling Stability**: DynaDiff demonstrates excellent sampling consistency (see our response to **Reviewer RMvc W4** for specific quantitative variance analysis).
> - **Weight Generation Baselines**: We added experimental comparisons with state-of-the-art weight-space learning methods (D2NWG ) and CVAE (see response to **Reviewer sf5a W1**).
>
> **W2 (Presentation):**
>
> - Regarding the specific construction process of the weight graph, please refer to Section 3.1.1 of the main text.
> - The training protocols, core settings, and hyperparameters for expert models are detailed in Appendices C and J.
>
> **W3&Q1 (Significance):**
>
> - **Offline/Online End-to-End Cost Decomposition**
>
>   Using Cylinder Flow ($96$ training environments, $400$ test environments) as an example, we compared total time costs on a single A100 GPU:
>
>   |          | Training cost (s) & GPU memory | Testing cost (s) & GPU memory |
>   | -------- | ------------------------------ | ----------------------------- |
>   | FNO      | 4225 (53G)                     | 828 (28G)                     |
>   | DPOT     | 11,392 (51G)                   | 1,855 (23G)                   |
>   | Poseidon | 3,334 (47G)                    | 1,634 (4.8G)                  |
>   | MPP      | 6,194 (63G)                    | 906 (4.4G)                    |
>   | DyAd     | 674 (2.9G)                     | 273 (2.3G)                    |
>   | LEADS    | 1,422 (5.1G)                   | 401 (2.3G)                    |
>   | CoDA     | 790 (3.8G)                     | 719 (3.8G)                    |
>   | GEPS     | 902 (3.3G)                     | 425 (2.1G)                    |
>   | CAMEL    | 772 (3.4G)                     | 310 (2.3G)                    |
>   | DynaDiff | 3,432 + 2,217 (6.6G)           | 629 (4.2G)                    |
>
>   DynaDiff's Zoo construction time ($3,432$s) is even less than the training time of some foundation models (e.g. DPOT, MPP), and the total training overhead is in the same order of magnitude as foundation models.
>
> - **Comparison of Inference Efficiency with Test-time Fine-tuning Methods**
>
>   We designed a challenge experiment: all meta-learning methods were allowed to perform test-time fine-tuning using $10$ observation frames until they reached performance comparable to DynaDiff (RMSE $\le 0.075$) at a learning rate of 1e-5
>
>   | Finetune until RMSE<=0.075     | DyAd | LEADS | CoDA | GEPS | CAMEL | DynaDiff |
>   | ------------------------------ | ---- | ----- | ---- | ---- | ----- | -------- |
>   | Finetuning cost (s)            | 1812 | 1903  | 2195 | 1858 | 1765  | ——       |
>   | Avg. Adaptation Time (s / env) | 4.53 | 4.76  | 5.49 | 4.65 | 4.41  | **1.57** |
>
>   Results indicate that existing adaptive methods require approximately **$200\%$** additional inference overhead to achieve comparable performance. This demonstrates DynaDiff's significant advantage in time-sensitive deployment scenarios.
>
> - **Generalization in Noisy Environments**
>
>   In experiments on the real-world weather dataset ERA5 (containing observation noise, Figure 4), DynaDiff's performance surpassed all baselines, proving its utility in complex, high-noise scientific computing scenarios.
>
> **W4 (Originality):**
>
> Generating complete weights is highly challenging, as minor weight distortions can lead to predictive collapse, requiring high-fidelity generation capabilities. We added comparative experiments with existing meta-learning/hypernetwork-style conditioning methods (see response to **Reviewer sf5a W1**). As shown in the figure ([weight_dist](https://anonymous.4open.science/r/icml-rebuttal-86BD/vis_weight.png)), we calculated the Jensen-Shannon Divergence between generated and true distributions. Simpler conditional generators (CVAE [1]) tend to produce blurred mean distributions (higher JSD), making it difficult to capture the multimodal nature of physical weights. Furthermore, compared to the similar diffusion weight generation baseline D2NWG [2] , DynaDiff's original weight graph and Dynamics-informed Prompter bring significant performance improvements.
>
> **Q2: Sampling Stability**
>
> DynaDiff's generation stability is excellent (with RMSE variance on the order of $10^{-3}$); see the analysis for **Reviewer RMvc W4**.
>
> ---
>
> [1] In-context meta LoRA generation
>
> [2] Diffusion-Based Neural Network Weights Generation

---

> > ### Author Rebuttal · Reviewer_pAWm · 2026-04-03
> >
> > Thanks authors for submitting rebuttal. I decided to keep the score.

---

> > > ### Author Response · Authors · 2026-04-04
> > >
> > > Thank you for confirming that our rebuttal has addressed your concerns. We appreciate your insightful comments, which have significantly helped us clarify the significance of our work.

---

### Official Review · Reviewer_RMvc · 2026-03-13

**Soundness:** 3
**Presentation:** 3
**Significance:** 3
**Originality:** 2
**Overall Recommendation:** 4
**Confidence:** 3

**Summary:**

The paper proposes DynaDiff, a generative framework designed to adapt physical dynamics models to varying unseen environments. Instead of traditional gradient-based fine-tuning, it introduces direct weight generation as an alternative paradigm. By pre-constructing a Model Zoo of expert weights and learning their joint distribution with environments via a Weight-space Diffusion model, the framework enables rapid adaptation to unseen environments using limited observations without any test-time optimization

**Compliance With Llm Reviewing Policy:**

Affirmed.

**Final Justification:**

I appreciate the effort to address my concerns, and I will raise my original score.

**Key Questions For Authors:**

1. In practical deployment, how should one select the final model among multiple stochastic generations? Is there a recommended validation metric or an ensemble strategy to mitigate variance?
2. Since all nodes are projected into the same dimension before attention, what specific signal prevents the MHA from confusing nodes across different layers?
3. Does the framework can detect when a prompt is in an extrapolation regime?

**Limitations:**

yes

**Strengths And Weaknesses:**

Strengths
- The paper introduces a novel adaptation method by treating model weights as a data modality, moving beyond traditional parameter modulation or iterative fine-tuning to directly generate the complete weight.
- Extensive experiments across various PDE systems demonstrate that DynaDiff outperforms competitive meta-learning and foundation model baselines.

Weaknesses
- There is a significant disparity between the size of the target model and the generator. While the predictive FNO model is lightweight at approximately 1M parameters, the DynaDiff generative module requires roughly 380M parameters. This raises concerns about the framework's scalability; if the target model were to scale to a larger neural operator, the generator's size might become computationally prohibitive.
- The Weight VAE employs multi-head attention, which is inherently permutation-invariant. Although the authors use layer-specific linear maps to protect node identity, there is a lack of rigorous analysis on whether the global attention mechanism can truly preserve the precise spatial and layer-wise ordering required for complex dynamics where the exact relative position of weights is functionally critical.
- While the ablation study shows that the two-stage (VAE+LDM) approach outperforms a single-stage direct diffusion model, the fundamental logical necessity for the iterative denoising process of Diffusion is not fully established against a simpler, well-optimized Conditional VAE (CVAE).
- As a stochastic process, Diffusion generates different weight sets for the same observation sequence. The paper focuses on average performance but lacks a quantitative analysis of performance variance. In safety-critical scientific applications, it is essential to understand the probability of generating outlier models that may fail to respect fundamental physical conservation laws.
- As indicated in the results, accuracy drops as environmental shifts move further away from the training distribution. It seems that there is a risk of the model "hallucinating" non-physical weights when faced with out-of-distribution (OOD) environments.

---

> ### Author Rebuttal · Authors · 2026-03-31
>
> We thank you for the constructive feedback and address your concerns as follow.
>
> **W1: On Generator Scale and Scalability**
>
> DynaDiff is computationally scalable. Taking an MLP predictor as an example (layers $L$, width $d$), total parameters are $O(d^L)$. Our weight graph design ensures the number of nodes and feature dimensions are $O(N \cdot L)$ and $O(d)$, respectively. Since the attention mechanism is independent of node count, the generator's parameter count (primarily projection layers) only changes **linearly** with $L$ and $d$, rather than exponentially with the predictor's parameter count. Our computational results also confirm this [(figure)](https://anonymous.4open.science/r/icml-rebuttal-86BD/params_scale.png).
>
> **W2 & Q2: Structural Fidelity and Layer Confusion**
>
> The feature vectors of nodes in the weight graph prevent MHA from confusing nodes across different layers. A node's feature vector consists of the connection weights from its lower layer; thus, spatial position and hierarchical connectivity are embedded during graph construction. We visualized the attention score distribution of the Weight VAE ([CF](https://anonymous.4open.science/r/icml-rebuttal-86BD/cy_attn_layers.png)、[LO](https://anonymous.4open.science/r/icml-rebuttal-86BD/lo_attn_layers.png) systems).  Attention weights exhibit distinct hierarchical structures, proving the global attention mechanism preserves precise hierarchical ordering.
>
> Additionally, we tested adding Sinusoidal Position Encoding (PE) to node features [1]. As shown below, PE provided no significant gain, further confirming that the original node features possess sufficient topological discriminability.
>
> | Reconstruction RMSE | CF         | LO          |
> | ------------------- | ---------- | ----------- |
> | Graph VAE + PE      | 0.060±0.04 | 0.074±0.011 |
> | Graph VAE           | 0.059±0.03 | 0.075±0.009 |
>
> **W3: Necessity of the Diffusion Model**
>
> - **Distribution Fitting**: We added **CVAE** as a baseline (see **Reviewer sf5a W1**). Experiments show that the iterative denoising of the diffusion process generates a distribution significantly closer to the true expert weight manifold than CVAE (error reduced by **more than 70%**, [figure](https://anonymous.4open.science/r/icml-rebuttal-86BD/vis_weight.png)).
> - **High-Fidelity Requirement**: Generating complete weights is challenging, as minor distortions can collapse predictive performance. The iterative denoising process allows the model to perform fine-grained adjustments in the latent space to match complex weight distributions [2].
>
> **W4 & Q1: Sampling Stability**
>
> Addressing concerns about outliers in safety-critical applications, we sampled 100 times across 10 OOD environments for CF and NS systems. DynaDiff's prediction variance was extremely low, and the worst-case RMSE outperformed the best baseline, indicating DynaDiff rarely samples outliers. This is likely because the Model Zoo, constructed via domain-adaptive initialization, creates smooth basins in the loss landscape.  Diffusion sampling occurs on a continuous manifold, ensuring a high performance floor.
>
> | RMSE          | CF           |       |       |      | NS          |       |       |       |
> | ------------- | ------------ | ----- | ----- | ---- | ----------- | ----- | ----- | ----- |
> |               | Mean         | Std   | Worst | Best | Mean        | Std   | Worst | Best  |
> | DynaDiff*100  | 0.061        | 0.002 | 0.68  | 0.56 | 0.063       | 0.003 | 0.067 | 0.058 |
> | Best baseline | 0.082 (GEPS) |       |       |      | 0.074 (FNO) |       |       |       |
>
> In deployment, for extreme stability, one can sample $k$ models and select the one with the minimum reconstruction residual on the observation sequence $X_L$.
>
> **W5: Extreme Distribution Shift**
>
> As shown in Appendix Table 6, while accuracy drops for all models in extreme extrapolation scenarios, DynaDiff’s performance decays the slowest and maintains a significant lead in the harshest conditions. This demonstrates that the paradigm of modeling the environment-weight manifold via weight generation is more robust than existing meta learning and foundation models.
>
> **Q3: Extrapolation Detection**
>
> Yes. DynaDiff can effectively detect when a prompt enters an extrapolation regime by leveraging the latent physical signatures captured by the Prompter.
>
> We developed a lightweight Random Forest detector using prompt vectors, with boundary environments (top 25% training error) as an extrapolation label. We evaluated it on the Cylinder Flow system:
>
> - Accuracy: 0.9441
> - Recall: 0.8571
> - Precision: 0.8824
>
> This confirms the Prompter distills discriminative physical information, allowing users to pre-evaluate model reliability before deployment ([figure](https://anonymous.4open.science/r/icml-rebuttal-86BD/cy_ood_detect.png)).
>
> ---
>
> [1] Diffusion-Based Neural Network Weights Generation
>
> [2] A survey of weight space learning: Understanding, representation, and generation

---

> > ### Author Rebuttal · Reviewer_RMvc · 2026-04-04
> >
> > I appreciate the effort to address my concerns, and I will raise my original score.

---

> > > ### Author Response · Authors · 2026-04-04
> > >
> > > Thank you for confirming that our responses have addressed the concerns. We appreciate the constructive feedback throughout this process.

---

### Decision · Program_Chairs · 2026-04-30

**Decision:**

Accept (regular)

**Comment:**

This paper proposes DynaDiff, a generative meta-learning framework for adapting dynamical system predictors under environmental shifts by directly generating model weights via a conditional diffusion process. It introduces a weight-graph representation, a functional loss to align model behavior, and a dynamics-informed prompter to condition generation from short observation sequences, achieving improved accuracy and efficient adaptation without test-time fine-tuning.

Reviewers found the problem important and the approach technically solid, with consistent empirical gains and a coherent design. Concerns focused on computational cost, necessity of diffusion, robustness, and evaluation breadth. The rebuttal addressed these with added experiments, analyses of scalability and variance, and stronger baselines; all reviewers acknowledged resolution of concerns and several raised scores. Given the strengthened evidence and positive consensus, I recommend accepting this paper.